# Clinical Study of Aspirin and Zileuton on Biomarkers of Tobacco-Related Carcinogenesis in Current Smokers

**DOI:** 10.3390/cancers14122893

**Published:** 2022-06-11

**Authors:** Linda L. Garland, José Guillen-Rodriguez, Chiu-Hsieh Hsu, Lisa E. Davis, Eva Szabo, Christopher R. Husted, Hanqiao Liu, Ashley LeClerc, Yuriy O. Alekseyev, Gang Liu, Julie E. Bauman, Avrum E. Spira, Jennifer Beane, Malgorzata Wojtowicz, H.-H. Sherry Chow

**Affiliations:** 1Department of Medicine, University of Arizona, Tucson, AZ 85724, USA; jebauman@gwu.edu; 2University of Arizona Cancer Center, University of Arizona, Tucson, AZ 85724, USA; jguillen@arizona.edu (J.G.-R.); pchhsu@arizona.edu (C.-H.H.); schow@arizona.edu (H.-H.S.C.); 3College of Pharmacy, University of Arizona, Tucson, AZ 85721, USA; ldavis@pharmacy.arizona.edu; 4Division of Cancer Prevention, National Cancer Institute, Bethesa, MD 20892, USA; szaboe@mail.nih.gov (E.S.); wojtowim@mail.nih.gov (M.W.); 5Section of Computational Biomedicine, Department of Medicine, School of Medicine, Boston University, Boston, MA 02118, USA; chusted@bu.edu (C.R.H.); hqliu@bu.edu (H.L.); gangl@bu.edu (G.L.); aspira@bu.edu (A.E.S.); jbeane@bu.edu (J.B.); 6Department of Pathology and Laboratory Medicine, School of Medicine, Boston University, Boston, MA 02118, USA; msrlab@bu.edu (A.L.); yurik@bu.edu (Y.O.A.); 7Division of Hematology/Oncology, Department of Medicine, George Washington (GW) University and GW Cancer Center, Washington, DC 20037, USA

**Keywords:** lung cancer, cancer prevention, aspirin, prostaglandin, zileuton, leukotriene

## Abstract

**Simple Summary:**

Aspirin and related drugs with anti-inflammatory effects have lung cancer prevention effects in laboratory studies and through their use in populations at risk for lung cancer. We studied aspirin plus zileuton compared to two placebo pills for 3 months in smokers. We studied genes associated with lung cancer, smoking, and lung disease. We used nasal swabs to collect nasal cells that were smoke exposed to look for changes in these genes. We found that most of the gene panels in the nasal cells related to smoking and lung cancer lung disease did not change in a favorable way, but we did see a favorable change in a gene panel representing abnormal squamous cells that may progress to lung cancer. We think this finding is of interest and can be studied further using deep lung biopsies to understand if this drug effect occurs where squamous cell lung cancer actually starts.

**Abstract:**

The chemopreventive effect of aspirin and other non-steroidal anti-inflammatory drugs (NSAIDs) on lung cancer risk is supported by epidemiologic and preclinical studies. Zileuton, a 5-lipoxygenase inhibitor, has additive activity with NSAIDs against tobacco carcinogenesis in preclinical models. We hypothesized that cyclooxygenase plus 5-lipoxygenase inhibition would be more effective than a placebo in modulating the nasal epithelium gene signatures of tobacco exposure and lung cancer. We conducted a randomized, double-blinded study of low-dose aspirin plus zileuton vs. double placebo in current smokers to compare the modulating effects on nasal gene expression and arachidonic acid metabolism. In total, 63 participants took aspirin 81 mg daily plus zileuton (Zyflo CR) 600 mg BID or the placebo for 12 weeks. Nasal brushes from the baseline, end-of-intervention, and one-week post intervention were profiled via microarray. Aspirin plus zilueton had minimal effects on the modulation of the nasal or bronchial gene expression signatures of smoking, lung cancer, and COPD but favorably modulated a bronchial gene expression signature of squamous dysplasia. Aspirin plus zileuton suppressed urinary leukotriene but not prostaglandin E2, suggesting shunting through the cyclooxygenase pathway when combined with 5-lipoxygenase inhibition. Continued investigation of leukotriene inhibitors is needed to confirm these findings, understand the long-term effects on the airway epithelium, and identify the safest, optimally dosed agents.

## 1. Introduction

Lung cancer is the leading cause of cancer-related deaths both in men and women in the US and worldwide. An estimated 228,820 new cases of lung cancer and 135,720 deaths due to lung cancer are projected to occur in the US in 2020. [1]. While the prevention of tobacco use and tobacco cessation efforts remain the most effective means to reduce the incidence of lung cancer, the risk of developing this disease is still elevated for years after successful quitting compared to never smokers. Moreover, relapse is high even with state-of-the art cessation methods. Effective and well-tolerated chemoprevention strategies for lung cancer are urgently needed.

Interest in aspirin (ASA) as a chemopreventive agent is based on epidemiologic cohort and case control studies showing a reduction in precancerous lesions and cancers in lung, colon, breast, and prostate tissue [2,3,4]. An analysis of eight randomized trials of daily ASA for a mean duration of ≥4 years versus no aspirin reported a significant decrease in lung adenocarcinoma mortality with ASA use (HR = 0.68, 95% CI 0.50–0.92) [5]. In a large cohort study (VITamin and Lifestyle Study), total NSAID use (>10 years) was associated with a borderline reduction in the risk of lung cancer (HR 0.82; 95% C.I. 0.64–1.04), with the strongest association for adenocarcinoma (HR 0.59); this effect was limited to men (HR 0.66) and to long-term (≥10 years) former smokers (HR 0.65) [5]. The effects of ASA are also supported in chemically induced murine models of lung cancer, in which it inhibits tumorigenesis and reduces pulmonary tumor multiplicity [6,7].

Zileuton is a potent inhibitor of 5-lipoxygenase (5-LOX) [8,9]. It has been approved by the FDA for the treatment of asthma based on the downstream suppression of leukotrienes, including leukotriene B(4) (LTB(4)), leukotriene C(4) (LTC(4)), leukotriene D(4) (LTE(4)), and leudotriene E(4) (LTE(4)), which are inflammatory mediators of asthma. 5-LOX and leukotrienes appear to be relevant therapeutic targets for lung chemoprevention based on the findings of higher levels of 5-LOX metabolites in a number of human solid tumors, including lung cancer and leukemias, compared to normal tissues [10]. Additional supporting evidence for the use of 5-LOX as a chemoprevention target is evidence that mRNA for 5-LOX and 5-LO activating protein (FLAP), which is required for activation of 5-LOX, is expressed in lung cancer cell lines and that mRNA for 5-LOX is expressed in lung cancer tissues [11].

In preclinical studies, 5-LOX inhibitors, including zileuton, have inhibitory activity in a number of lung cancer models [12,13]. In an A/J mouse lung chemoprevention model using vinyl carbamate induction, the administration of zileuton 1200 mg/kg in the diet (equivalent to a clinically relevant human dose) starting 2 weeks after carcinogen exposure caused a significant reduction in tumor multiplicity (24% at 13 weeks and 28% at 43 weeks) and reduced the size of lung tumors [14].

The combination of ASA and zileuton may be additive or synergistic in inhibiting the arachidonic acid (AA) pathway of inflammatory mediators related to lung carcinogenesis. In an NNK-induced model of lung carcinogenesis, ASA plus the 5-LOX inhibitor A-79175 was more effective than either drug alone, suggesting that concurrent inhibition of both 5-LOX and cyclooxygenase (COX) is more effective than inhibition of either pathway alone [12]. A study of the combination of zileuton and the selective COX-2 inhibitor celecoxib in smokers reported a significant reduction in the levels of prostaglandin E metabolite (PGE-M) and LTE(4), suggesting that the combination resulted in inhibition of both the LOX and COX proinflammatory pathways of AA metabolism [15]. The combination of celecoxib with zileuton led to a 62% reduction in PGE-M levels compared to an 18% reduction in PGE-M with zileuton alone; furthermore, the addition of celecoxib to zileuton did not affect the inhibition of LTE(4) by zileuton alone.

Exposure to cigarette smoke injures the cells that line the respiratory tract and creates an airway field of injury. Gene expression alterations in the airway field have been identified that are associated with smoking [16,17], smoking cessation [18,19], COPD [20,21], bronchial premalignant lesions [22,23,24], and lung cancer [25]. There is significant overlap between bronchial and nasal smoking- and lung cancer-associated gene expression changes [26,27], suggesting the ability to detect lung disease-related biology throughout the intra- and extra-thoracic airway. We previously reported on the effect of a low-dose (LD) ASA intervention for 3 months on gene expression in the nasal epithelium in current smokers [28]. The ASA intervention had minimal effects on known carcinogenesis gene signatures in the nasal epithelium. However, gene set enrichment analysis (GSEA) showed that the ASA intervention led to an enrichment of genes in pathways related to the biology of ribosomes, histones, proteasomes, chemokines, the mitochondrial electron transport chain, cellular signaling, and immune function.

Given the efficacy of combined COX- and 5-LOX inhibition in preclinical lung carcinogenesis models and human studies showing that dual COX and LOX inhibition decreased the respective metabolites of the AA pathway in smokers, we sought to evaluate the effects of LD ASA and zileuton on the airway field of injury via gene expression profiling of nasal brushings. Here, we present a study of current smokers randomized to a 12-week intervention of ASA and zileuton vs. double placebo (NCT02348203) with gene expression from nasal brushings profiled at baseline, at the end of the intervention, and at one-week post intervention.

## 2. Materials and Methods

### 2.1. Study Design

This study was a single-center, randomized, double-blinded, placebo-controlled trial to determine the modulatory effects of combined treatment of ASA and zileuton on nasal epithelium gene expression and AA metabolism in current smokers. The primary endpoint was changes in a smoking-associated gene expression signature derived using bronchial and nasal brushings (*n* = 119 genes) [29] in the combined ASA and zileuton group vs. placebo control.

Secondary endpoints included: (1) the effects of the treatment with ASA and zileuton vs. placebo on the modulation of additional gene expression signatures, including: a nasal lung cancer signature (*n* = 535 genes; abbreviated signature *n* = 35 genes) [29]; a bronchial smoking signature (*n* = 81 rapidly reversible genes upon smoking cessation) [30]; a bronchial lung cancer gene signature (*n* = 23 genes) [31]; a PI3K pathway activity signature observed in bronchial brushings from smokers and lung cancer patients (*n* = 183 genes) [23]; a bronchial COPD signature (*n* = 98 genes) [20]; a bronchial signature associated with the presence of squamous dysplasia (*n* = 280 genes) [22]; and a signature associated with premalignant lesion severity derived from endobronchial biopsies (*n* = 200 genes) [24]; (2) persistence of the changes in the smoking gene expression signature in the nasal epithelium one week off agent intervention; (3) changes in cyclooxygenase (COX) and 5-lipoxygenase (LOX)-mediated arachidonic acid (AA) metabolism; (4) safety of 12-week exposure to ASA and zileuton vs. placebo in current smokers; (5) exploratory analysis of a gender effect in the modulatory effects of ASA and zileuton on the smoking-related gene expression signature; (6) modulation of the metabolomics profile of the AA pathway by ASA and zileuton; and (7) unbiased discovery of the effects of ASA and zileuton on whole-genome gene expression.

### 2.2. Study Drugs

ASA 81 mg and matched placebo were provided by the National Cancer Institute, Division of Cancer Prevention (NCI DCP) and packaged and supplied to the study site by the NCI DCP Drug Repository, MRIGlobal, Kansas City, MO. Zileuton (Zyflo CR™, Cary, NC, USA) 600 mg was purchased from commercial sources and repackaged by the University of Arizona Cancer Center Investigational Pharmacy in bottles supplied by the NCI DCP Drug Repository. A not fully matched zileuton placebo was packaged and supplied by NCI DCP Drug Repository.

### 2.3. Study Population

Current smokers at least 18 years of age with a ≥20 pack year tobacco exposure history and an average daily use of ≥10 cigarettes per day were recruited from the greater Tucson and Phoenix areas. Inclusion criteria included normal hematologic, biochemical, and coagulation parameters and an ability to participate in the trial and sign informed consent. Exclusion criteria included: allergy to aspirin or NSAIDs; gastric intolerance to ASA or NSAIDs; history of gastric ulcer; ASA or NSAID use for more than 5 days per month within 3 months of enrollment; unwilling or unable to refrain from use of non-study ASA or NSAID; adult asthma; current, recent, or chronic use of leukotriene antagonists or glucocorticoids (systemic, topical and/or nasal sprays); requiring chronic anticoagulation or anti-platelet therapy; history of a bleeding disorder or hemorrhagic stroke; history of chronic sinusitis or recent nasal polyps; history of, or current, active or chronic liver disease; unwilling or unable to limit alcohol consumption; pregnant or lactating; inability to absorb an oral agent; uncontrolled intercurrent illness; invasive cancer within five years except non-melanoma skin cancer; or taking drugs known to interact with zileuton, including theophylline, warfarin, and propranolol.

### 2.4. Study Procedure

Participants underwent a physical exam, clinical laboratory analysis (CBC and CMP), and assessments of medical history, concurrent medications, NSAIDs use, and tobacco use history at the eligibility evaluation. Participants who had taken NSAIDs within the preceding 2 weeks underwent a 4-week washout period before baseline specimen collection. Participants underwent baseline specimen collection of nasal brushing, urine, blood, and buccal cells. Participants were then randomized (1:1) to receive ASA (81 mg QD) and zileuton (Zyflo CR™) two 600 mg extended-release tablets BID or placebo pills for 12 weeks. For study visit scheduling conflicts, the agent intervention was extended for 2 weeks until the rescheduled visit. Two interim study visits were conducted at 4 and 8 weeks for hepatic function testing, compliance check, AE review, and a current tobacco use assessment. Following the agent intervention, an end-of-intervention visit was conducted for safety labs (CBC and CMP), compliance check, current tobacco use assessment, and collection of nasal brushing, urine, blood, and buccal cells. Biospecimen collection was also performed 7–10 days post-intervention visits. The safety of the agent intervention was assessed by self-reported AEs and clinical laboratory analysis. AEs were graded using the NCI Common Terminology Criteria for Adverse Events (CTCAE) v. 4.0. Upon study completion, participants were provided information on the Arizona Smokers’ Helpline to assist in smoking cessation.

### 2.5. Nasal Brushing for Gene Expression Analysis

Nasal epithelium brushings were collected using a nasal speculum to spread one nostril while a standard cytology brush was inserted underneath the inferior nasal turbinate. The brush was rotated in place for 3 s and immediately placed in RNAProtect Cell solution. A second brushing from the same nostril was similarly collected and processed. Samples were stored at −80 °C prior to analysis.

### 2.6. Analysis of Urinary Biomarkers of Arachidonic Acid Metabolism

Prostaglandin metabolite E_2_ (PGE_2_) is a major COX-mediated AA metabolism product. The major urinary metabolite of PGE_2_, 11α-hydroxy-9,15,-dioxo-2,3,4,5-tetranor-prostane-1,20-dioic acid (PGEM), was quantified by a sensitive and specific liquid chromatography tandem mass spectrometry (LC-MS) assay as previously described [32].

The urinary LTE4, the terminal product of 5-LOX-mediated AA metabolism, was quantified by a sensitive and specific liquid chromatography tandem mass spectrometry assay [33]. Briefly, 5 mL urine was acidified to pH 3 with 1M HCl and mixed with the internal standard ([^2^H_3_]LTE4 (1 ng)) and extracted with C_18_ solid-phase extraction columns. The eluate from solid-phase extraction was dried and reconstituted in an aliquot of methanol and filtered using a 0.2 µm Spin-X filter. The filtrate was dried and reconstituted in an aliquot of methanol/water (50/50) prior to injection onto the LC-MS system. The chromatographic separation was achieved by a C_18_ reverse-phase column and a gradient mobile phase of ammonium acetate, acetic acid, and acetonitrile. The mass spectrometer was operated in negative ion mode utilizing electrospray ionization. Detection was achieved using selected reaction monitoring, with the transition of m/z 438 to 333 monitored for LTE4 and the transition of m/z 441 to 336 for the internal standard. The assay was linear over the range of 6.25–2500 pg/mL with an assay accuracy of >90% and an inter-assay coefficient of variation of <12%.

Urinary biomarker levels were normalized by urinary creatinine concentrations measured using a creatinine assay kit (Diazyme Laboratories).

### 2.7. Microarray Data Acquisition and Data Preprocessing

Total RNA was isolated from nasal brushings using a Qiagen miRNeasy Mini Kit following the manufacturer’s instruction. Quality control and quantification of the RNA samples were performed using an Agilent BioAnalyzer and NanoDrop spectrophotometer. The total RNA was processed and hybridized to Affymetrix Human Gene 1.0 ST Arrays.

Gene expression values were generated for each Human Gene Arrays 1.0ST CEL file (*n* = 127 samples, *n* = 43 subjects) using R statistical software (version 3.6.0) and the Robust Multiarray Average (RMA) algorithm (affy package) [34] with Entrez Gene-specific probeset mapping (version 23.0.0) from the Molecular and Behavioral Neuroscience Institute (Brainarray) at the University of Michigan [35]. Standardized RNA quality metrics were assessed, including the normalized un-scaled standard error (NUSE, cutoff > 1.05) and relative log expression (RLE, cutoff > 0.1). Additionally, we conducted a principal component analysis (PCA) across all genes and samples and excluded samples that were greater than 2 standard deviations from the mean of the first principal component. Samples with more than one failed quality metric were excluded from analysis (*n* = 4). limma [36] was used to remove the effect of the variable RNA quality based on RNA integrity (RIN) values. The sex annotation of each sample was verified using the expression levels of Y-chromosome specific genes.

### 2.8. Calculation of Gene Expression Signature Scores

For each previously published gene expression signature (“signature data”), the corresponding processed gene expression data used to derive the signatures was downloaded from the Gene Expression Omnibus (GEO). Specifically, we downloaded the following datasets: GSE16008 for the smoking-associated gene expression signature derived from nasal and bronchial brushings, GSE80796 for the lung cancer-associated gene expression signature derived from nasal brushings, GSE7895 for the smoking-associated gene expression signature derived from bronchial brushings, GSE12815 for the PI3K activity signature, GSE37147 for the COPD-associated gene expression signature derived from bronchial brushings, GSE79209 for the squamous dysplasia-associated signature derived from bronchial brushings, and GSE109743 for the proliferative molecular subtype signature derived from endobronchial biopsies and reflected in the bronchial brushings. For GSE79209 and GSE109743 (endobronchial biopsies in the discovery cohort (*n* = 190)), residual gene expression values, as previously described [22,30], were used in this analysis. For each gene expression dataset (“signature data”), ComBat [37] was used to remove batch effects between the signature data and the study gene expression data. As noted above, prior to conducting ComBat, the effect of RIN was removed from the study gene expression data prior to performing ComBat with the signature datasets. However, prior to performing ComBat with the GSE79209 and GSE109743 signature datasets, residual gene expression values were calculated in the study dataset using limma, adjusting for batch and RIN. ComBat-adjusted gene expression values were z-score normalized across the combined study data and signature data. For each gene signature, principal component analysis was conducted across the signature data, and the first principal component was applied to the study data to generate gene signature scores (Appendix A). Additionally, we generated scores from a lung cancer-associated gene expression signature derived from bronchial brushings using the classifier described by Whitney et al. [31] to score the study data.

### 2.9. Identification of Gene Expression Changes Associated with Aspirin and Zileuton Treatment

A linear mixed effect model implemented in limma was used to identify differentially expressed genes associated with ASA plus zileuton at the end-of-treatment or one-week post-treatment points. The main effects in the model were treatment (ASA plus zileuton or placebo), timepoint (baseline, end-of-treatment, one-week post-treatment), the interaction between treatment and timepoint, RIN, and batch (samples were processed in 3 batches). Patient was modeled as a random effect using the limma ‘duplicateCorrelation’ function. Genes signatures associated with ASA plus zileuton at the end-of-treatment and one-week post-treatment points were identified based on empirical Bayes moderated *t*-statistics and their associated *p*-values for the interaction terms (*p*-value < 0.01, via the limma topTable function). The relationship between the differentially expressed genes with ASA plus zileuton treatment at the end-of-treatment and one-week post-treatment points was assessed using Gene Set Enrichment Analysis (GSEA) [38] using ranked gene lists based on the moderated t-statistic and gene signatures for the interaction effects (FDR < 0.05). Additionally, the ranked lists for each interaction effect were used for pathway enrichment analysis via GSEA using the Molecular Signatures Database (MSigDB) [39] hallmark gene set (FDR < 0.25). Finally, we evaluated the correlation (Spearman) between gene scores computed using Gene Set Variation Analysis (GSVA) [40] for each gene signature (ASA plus zileuton at end-of-treatment and one-week post-treatment up- and downregulated genes) and PGEM, LTE4, and cotinine analyte measurements. Correlation metrics were computed using changes in the gene scores and analyte measurements for each patient between the end-of-treatment and baseline or one-week post-treatment and baseline points, and correlation metrics were computed separately for placebo and drug arms across subjects with samples from all three timepoints (*n* = 38 subjects, *n* = 114 samples).

### 2.10. Statistical Methods

For a given endpoint, evaluable participants were defined as those that received the intervention and had three time points available for deriving changes within the same subject and then comparing the changes between the two groups. A two-sided two-sample *t*-test was used to test for significant differences in changes in the gene signature scores between the treatment and placebo groups. Based on a sample size of 20 per group, the power will be at least 85% to detect an effect size of ≥1 (i.e., ≥1 standard deviation difference in the mean changes between the 2 groups) at a significance level of 5%. A two-sided two-sample *t*-test was also used to compare the baseline values of the gene signature scores between the intervention arms and the baseline values of PGEM and LTE4 and changes in the PGEM and LTE4 levels between the intervention arms. A two-sided paired *t*-test was performed to evaluate the changes in the gene signature scores, PGEM, and LTE4 overall, by intervention arm, and by gender. All secondary analyses are considered exploratory so no correction for multiple comparisons was used. The Fisher’s exact test was used to compare the frequency of adverse events (AEs) between the intervention and placebo arms.

The protocol is available at: https://clinicaltrials.gov/ProvidedDocs/03/NCT02348203/Prot_SAP_000.pdf (accessed on 19 April 2022).

## 3. Results

### 3.1. Participant Demographics

From January 2016 to December 2018, 123 participants were consented and 63 enrolled and randomized to ASA and zileuton (*n* = 31; 18 male/13 female) or placebo (*n* = 32; 18 male/14 female) (see consort diagram, Figure 1).

In total, 21 (68%) participants in the combined ASA and zileuton arm and 22 (69%) in the placebo arm completed the study intervention with complete sets of biospecimens. The baseline characteristics of the randomized participants are summarized in Table 1.

In total, 21 (68%) participants in the combined ASA and zileuton arm and 22 (69%) in the placebo arm completed the study intervention with complete sets of biospecimens. The baseline characteristics of the randomized participants are summarized in Table 2.

Nasal specimens from 19 (61%) and 21 (66%) participants in the combined ASA and zileuton arm and placebo arm, respectively, yielded high-quality RNA and microarray data across all three time points (baseline, end-of-intervention, and post-intervention) and thus constituted the evaluable cohort for changes in the gene signature scores. The baseline characteristics of the cohort evaluable for gene signature scores are summarized in Appendix A. The majority of participants were white non-Hispanic; 11% of all participants were Hispanic. The mean age of all randomized participants was 50 and 54 years, respectively, for the ASA and zileuton and placebo arms; there was a significant difference in age (50 vs. 56 years, respectively, *p* = 0.03) between the intervention arms for participants with gene signatures. There were no significant differences by the intervention arm for body mass index (BMI), pack year tobacco exposure, or race/ethnicity for all participants or for those with evaluable gene signatures.

All but three participants had high baseline urinary cotinine levels consistent with self-reported current heavy tobacco use, with similar levels between arms and which remained high over the duration of the study (data not shown).

### 3.2. Adherence and Safety

Participants in both arms took greater than 90% of the assigned pills on average by pill count. The ASA plus zileuton intervention was overall well tolerated. All adverse events (AEs) were grade 1 or 2 events treated with supportive care intervention and were self-limited (Table 3). There were no AEs related to GI bleed, a well-known side effect of ASA, reported for the ASA and zileuton arm, although one event of anemia was reported for the placebo arm. Dyspepsia was more frequent in the ASA and zileuton arm than the placebo arm, as were other GI-related AEs (nausea, vomiting, diarrhea). Abnormal liver tests were very rare for both arms and not significantly different. AEs of headache, mania, fatigue, and rash were noted for ASA and zileuton, most likely related to zileuton.

The attrition rate for the ASA and zileuton arm was 10/31 and 10/32 for the placebo arm; reasons included lost to follow-up, withdrew consent, non-compliance, and AE. The two AEs that were associated with participant withdrawal from the study were abdominal pain grade 2 (1 participant, placebo arm; attribution possibly related to intervention) and seizure grade 2 (1 participant with history of seizures, ASA + zileuton arm; attribution unrelated to intervention).

### 3.3. ASA and Zileuton Do Not Modulate the Smoking-Associated Gene Expression Signature; However, They Do Modulate a Bronchial Squamous Dysplasia Signature

Gene expression data from the nasal brushings of participants in the ASA and zileuton and placebo arms was scored based on several lung-associated gene expression signatures, including signatures associated with smoking, lung cancer, COPD, and bronchial dysplasia, derived from bronchial and nasal brushings and endobronchial biopsies (see Appendix A). Participants with high-quality microarray data across all three time points (baseline, end-of-intervention, and post-intervention) were used to evaluate changes in the signature scores associated with ASA plus zileuton (*n* = 19 participants in the ASA and zileuton arm and 21 participants in the placebo arm). Data for the primary endpoint analysis of modulation of the smoking-associated gene expression signature score derived from nasal and bronchial brushings [29] are summarized in Table 4. 

There was no significant change in this gene signature score at the end of the intervention compared to baseline in the ASA and zileuton versus placebo groups (change of −0.15 ± 2.89 vs. 0.26 ± 2.26, respectively, *p* = 0.26). Scores derived from a bronchial signature associated with the presence of squamous dysplasia [22], a 280-gene set with upregulated genes enriched for energy metabolism (OXPHOS, the electron transport chain, and mitochondrial protein transport), are summarized in Table 5.

There was significant modulation of this signature score in a favorable direction from baseline to the end of the intervention with combined ASA and zileuton compared to placebo (change of −2.31 ± 4.08 vs. 1.94 ± 4.28, respectively, *p* < 0.01) (Figure 2). Data for the modulation of the other gene expression signature scores are summarized in Appendix A. No other gene expression signature was significantly modulated by the intervention vs. placebo.

### 3.4. ASA and Zileuton Suppresses the 5-LOX-Mediated AA Metabolite LTE4 but Does Not Suppress the COX-Mediated AA Metabolite PGEM

Urinary PGEM and LTE4 levels were analyzed for the 43 enrolled participants, including the 40 participants with evaluable gene expression data. The combined ASA and zileuton intervention significantly suppressed urinary LTE4 levels compared with the placebo arm, with a change of −57.62 ± 65.56 vs. 35.17 ± 92.67 pg/mg Cr, respectively, *p* < 0.001 (Table 6; Figure 3). 

A gender effect was noted in that a significant suppression of LTE4 was seen for females only (*p* < 0.001). LTE4 baseline levels were higher for females than males for both treatment arms. For the combined ASA and zileuton intervention, PGEM levels decreased from baseline to the end of the intervention, but this was not statistically significant (Appendix A). From the end-of-intervention to one-week post agent time points, there were non-significant increases in PGEM and LTE4 towards baseline values.

### 3.5. Aspirin and Zileuton Modulate Nasal Gene Expression

We identified 83 and 139 differentially expressed genes in the nasal epithelium at the end of the ASA plus zileuton treatment and one week after the ASA plus zileuton treatment, respectively, compared to baseline (*p* < 0.01) (Figure 4a,b). For each gene signature, principal component analysis was conducted across the signature data, and the first principal component was applied to the study data to generate gene signature scores (Appendix A).

(a)In total, 83 genes were associated with the end of the aspirin and zileuton treatment compared with baseline (*p* < 0.01). The heatmap shows hierarchal clustering of the z-score-normalized change in the gene expression between the end-of-treatment and baseline time points for each subject in the aspirin and zileuton (green) or placebo (purple) arms (*n* = 38 subjects).(b)In total, 139 genes were associated with the one-week post aspirin and zileuton treatment time point compared with baseline (*p* < 0.01). The heatmap shows hierarchal clustering of the z-score-normalized change in the gene expression between the one-week post-treatment and baseline time points for each subject in the aspirin and zileuton (light green) or placebo (light purple) arms (*n* = 38 subjects).(c)Genes that were altered at the end of the aspirin and zileuton treatment compared with baseline are concordant with the genes that were altered one week after the aspirin and zileuton treatment by GSEA (FDR < 0.05), suggesting that changes persist for at least one week post-intervention. Genes were ranked by the moderated t-statistic for the effect of the end of the aspirin and zileuton treatment versus baseline (*x*-axis) and the gene sets represent the genes that were up- and downregulated one week after the aspirin and zileuton treatment from (b). The black vertical lines represent the position of the genes in the ranked list (*x*-axis) and the height corresponds to the magnitude of the running enrichment score form GSEA (*y*-axis).(d)Ranked lists (genes ranked by moderated t-statistics) for the end-of-treatment and one-week post-aspirin and zileuton treatment versus baseline effects were used for pathway enrichment analysis via GSEA using the MSigDB hallmark gene set. The scatterplot shows the normalized enrichment scores for pathways enriched (FDR < 0.25) at the end of (*x*-axis) and one week after (*y*-axis) the aspirin and zileuton treatment. Positive and negative scores represent pathways enriched among upregulated and downregulated genes with the end of (yellow) and one week after (blue) or both (green) aspirin and zileuton treatment, respectively.

Gene expression alterations associated with the ASA plus zileuton treatment were relatively weak; however, up- and downregulated genes associated with the one-week post ASA plus zileuton treatment were significantly and concordantly enriched among genes associated with the end of treatment ASA plus zileuton (false discovery rate (FDR) < 0.05, Figure 4c).

Interestingly, pathway analysis also indicated an overlap between the genes that were modulated at the end of ASA plus zileuton treatment (*n* = 12 pathways, FDR < 0.25) and one week after the ASA plus zileuton treatment (*n* = 29 pathways, FDR < 0.25). Pathways associated with oxidative phosphorylation, reactive oxygen species, MYC targets, DNA repair, and MTORC1 signaling were enriched among downregulated genes and KRAS signaling and myogenesis were enriched among upregulated genes with the ASA plus zileuton end-of-treatment and one-week post-treatment time points (Figure 4d; Appendix A). The downregulation of oxidative phosphorylation genes with the ASA plus zileuton treatment is in line with the results above showing modulation of the squamous dysplasia signature, which is enriched for genes in this pathway [22]. Immune-associated pathways, including complement and IL2/STAT5 signaling, were downregulated one week after the ASA plus zileuton treatment. We did not identify significant correlations between changes in the gene signature score and changes in the LTE4, PGEM, or cotinine analytes between the baseline and end-of-treatment or one-week post-treatment time points and changes in either the placebo or drug study arms (*p* > 0.05).

## 4. Discussion

We investigated the effect of 12 weeks of combined ASA and zileuton versus placebo in current heavy smokers on nasal gene expression. We tested whether a comprehensive set of nasal and bronchial epithelium-derived gene expression signatures associated with smoking, lung cancer, COPD, and bronchial dysplasia were altered by the intervention. The ASA and zileuton intervention did not significantly modulate smoking, lung cancer, and COPD gene expression signatures but did favorably modulate one of two bronchial gene expression signatures for squamous dysplasia, in the direction of the non-dysplastic bronchial epithelium. This bronchial squamous dysplasia signature [22] was derived using normal-appearing bronchial brushes collected from subjects with or without bronchial dysplasia and is enriched with genes involved in oxidative phosphorylation that were upregulated in subjects with bronchial dysplasia. Changes in the signature over time were also shown to be associated with the progression/persistence versus regression of bronchial dysplasia. Analysis of the differentially expressed genes between baseline and either the end of the intervention or one-week post-intervention detected relatively few alterations; however, we observed that the gene expression changes associated with the intervention persisted at one week post-intervention. GSEA showed downregulation of pathways relevant to carcinogenesis and oxidative phosphorylation, corroborating the findings on the squamous dysplasia signature, which is enriched for genes in this pathway. While the squamous dysplasia signature has yet to be validated in the nasal epithelium, this signature’s modulation by combined COX- and 5-LOX inhibition is of high interest. Further study, however, is needed to determine if the bronchial squamous dysplasia signature could be modulated by other agents as the combination of COX- and 5-LOX or zileuton 5-LOX inhibitor is challenging for long-term use, given the complex dosing schedule of zileuton and rare neurotoxicity in otherwise healthy current or former smoker populations. The results indicate that the effect of the intervention persists after one week, so intermittent dosing may be possible. It would also be of interest to evaluate other leukotriene inhibitors with more favorable safely profiles in the modulation of this signature.

The selection of low-dose rather than regular-strength ASA in combination with zileuton, and the relatively short duration of the intervention may have led to the minimal effects observed on the pre-selected gene expression signatures. While a body of data supports the development of LD ASA in lung cancer chemoprevention [36,41], it is possible that regular-strength ASA might have yielded more robust modulation of the gene signatures in the anticarcinogenic direction. An additional arm of regular-strength ASA would have strengthened this study by addressing the dose–response in modulating gene expression associated with tobacco exposure and lung cancer. Additionally, the use of *fixed* low-dose ASA may have influenced the modulation of the gene expression signatures, as a recent analysis of ASA primary prevention trials of cardiovascular events and secondary prevention of colorectal cancer showed an interaction between LD ASA effects and body size, with the LD ASA effect seen only in participants <70 kg [42]. This study’s small sample size precludes a well-powered analysis to identify a correlation between body size and changes in gene signature scores unless the effect were to be strikingly large.

The optimal duration of combined COX and 5-LOX inhibitors to modulate tobacco- and lung cancer-related gene expression in the nasal epithelium as a surrogate for the respiratory epithelium is unknown. This study’s relatively short (12 week) intervention is broadly consistent with a number of preclinical biomarker studies of NSAIDs and other classes of chemoprevention agents in murine models of cigarette smoke exposure [43,44]. While murine models may not adequately represent the pharmacologic modulation of human chronically smoke-exposed respiratory epithelium, as murine models require high doses of both the carcinogen and chemopreventive agent to detect an effect in a short period of time, our previously reported study of low-dose ASA in a similarly designed study of a 12-week intervention showed modulation of gene expression signatures over this duration of the intervention [28]. Future studies that include cohorts exposed to longer durations of the intervention could provide valuable data on optimal dosing strategies.

The combined ASA and zileuton intervention significantly suppressed AA 5-LOX-derived metabolite urinary LTE4 levels compared with the placebo arm in females only. While a study of zileuton pharmacokinetics showed that gender effects on the pharmacokinetics were absent after a correction for bodyweight differences [45], an effect of androgens in impeding the biosynthetic 5-LO/FLAP complex assembly required for leukotriene synthesis has been reported for rodent models and this gender difference in bioactive 5-LO/FLAP complexes was noted in human leucocytes [46]. The combined ASA and zileuton did not, however, suppress AA COX-mediated urinary PGEM metabolite levels. It is recognized that there is complex crosstalk between the three branches of AA metabolism (COX, LOX, and cytochrome P450 (CYP)) [47]. The inhibition of single or multiple branches may result in the shunting of AA metabolism through other branches. In a human study, the administration of a selective COX-2 inhibitor caused shunting of AA metabolism into the proinflammatory LOX pathway in smokers with high baseline levels of urinary PGE-M. We showed that combined LD ASA and zileuton was effective in suppressing 5-LOX but not COX-mediated AA metabolism in high-risk smokers. It is plausible that LD ASA is not sufficient to suppress the COX-mediated AA metabolism due to potential shunting to the COX pathway when combined with a potent 5-LOX inhibitor.

LD ASA and zileuton was, in general safe, and well tolerated, with minimal GI toxicity and no evidence of GI bleeding, in keeping with studies using the NSAID naproxen, which showed that combined blockade of COX and 5-LOX did not produce additive GI AEs typically associated with naproxen administration, and reports of reduced GI toxicity associated with dual COX/5-LOX inhibitors such as licofelone [48]. There was a low frequency of zileuton-mediated grade 1–2 neurologic and psychiatric toxicities. A high attrition rate of around 33% in both arms was higher than that observed in our prior studies in this population and may be in part due to the requirement of 5 pills daily and BID dosing, which makes further development of this particular COX/LOX combination in the chemoprevention setting unlikely.

A weakness in the study design is the lack of quality of life (QOL) endpoints, which could elucidate the impact of this ASA plus zileuton strategy on QOL and thus guide similar chemoprevention strategies with the goal of, at the most, a modest impact on QOL in an at-risk but otherwise healthy population. A second weakness in the study design is the inability to better inform the optimal duration of the intervention to modulate the study endpoints, given that the single duration intervention was short (3 months). Additional cohorts utilizing longer durations of intervention (i.e., 6 or 12 months) would have strengthened the study.

## 5. Conclusions

Our study showed that LD ASA and zileuton compared to placebo modulated a bronchial squamous dysplasia gene signature but had no significant effects on other known carcinogenesis gene signatures in the nasal epithelium of heavy smokers. Inhibition of the leukotriene pathway remains of interest in the chemoprevention of squamous PMLs. Nasal gene expression signature determination is a novel approach to biomarker analysis, giving an approximation of the pulmonary milieu without the requirement of invasive tissue sampling. Nasal brushing of the nasal epithelium was well tolerated in this study and offers the potential to perform mechanistic lung cancer prevention studies in a less invasive way. This approach should be further explored with concomitant bronchial biopsies to determine if modulation of the squamous dysplasia gene signature can replace bronchoscopic biopsy for endpoint determination in future chemoprevention trials.

## Figures and Tables

**Figure 1 cancers-14-02893-f001:**
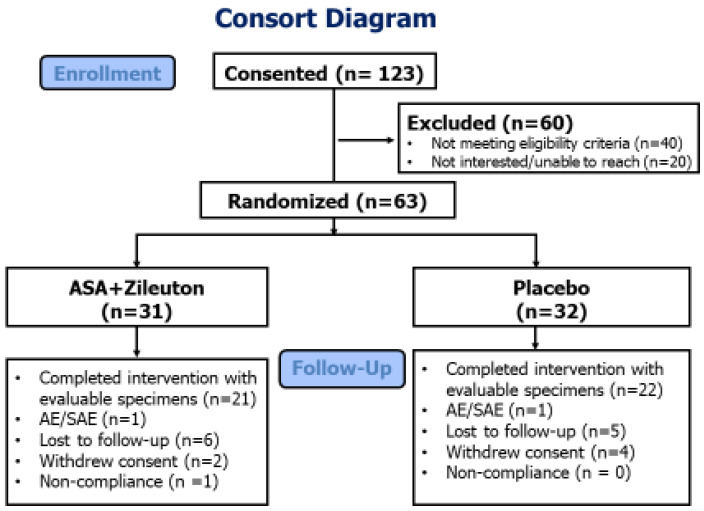
Consort flow diagram.

**Figure 2 cancers-14-02893-f002:**
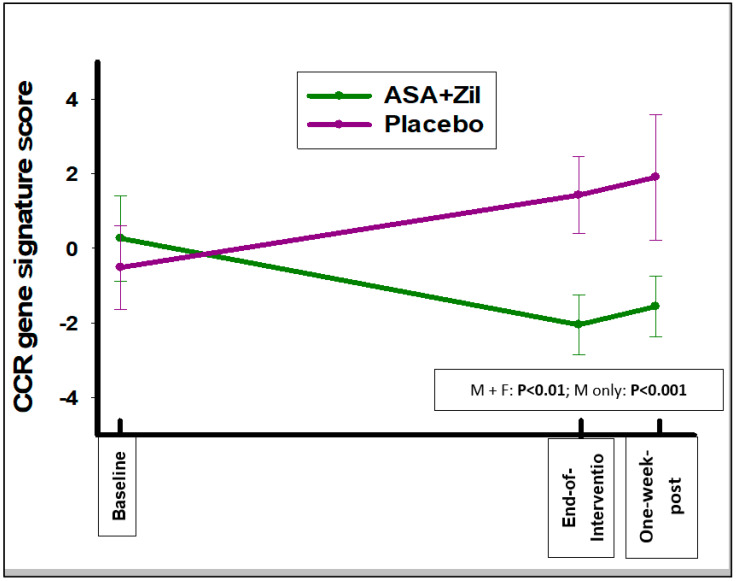
Bronchial dysplasia signature score at the baseline, end-of-intervention, and one-week post-intervention time points (pos→neg z-score is in the favorable direction) [22].

**Figure 3 cancers-14-02893-f003:**
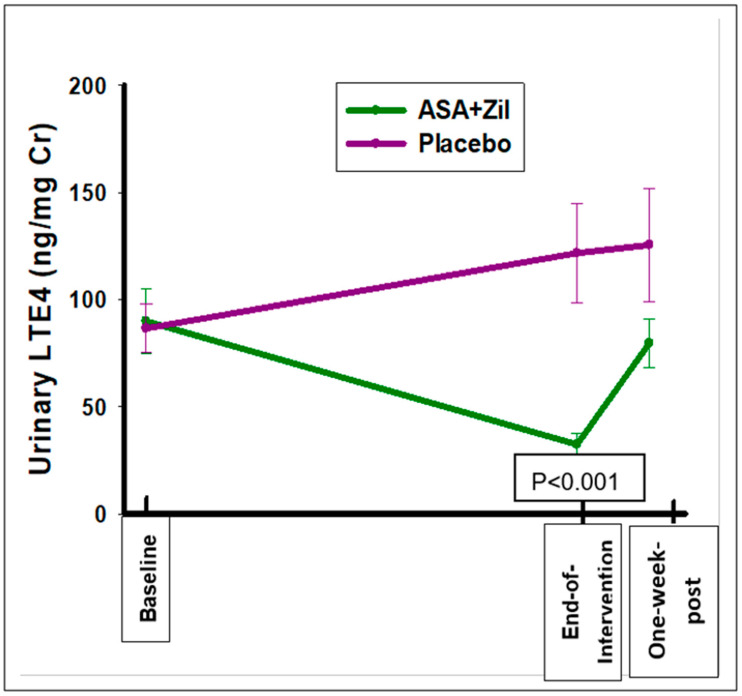
Baseline and changes in urinary LTE4 levels at baseline, end-of-intervention, and one-week post-intervention time points.

**Figure 4 cancers-14-02893-f004:**
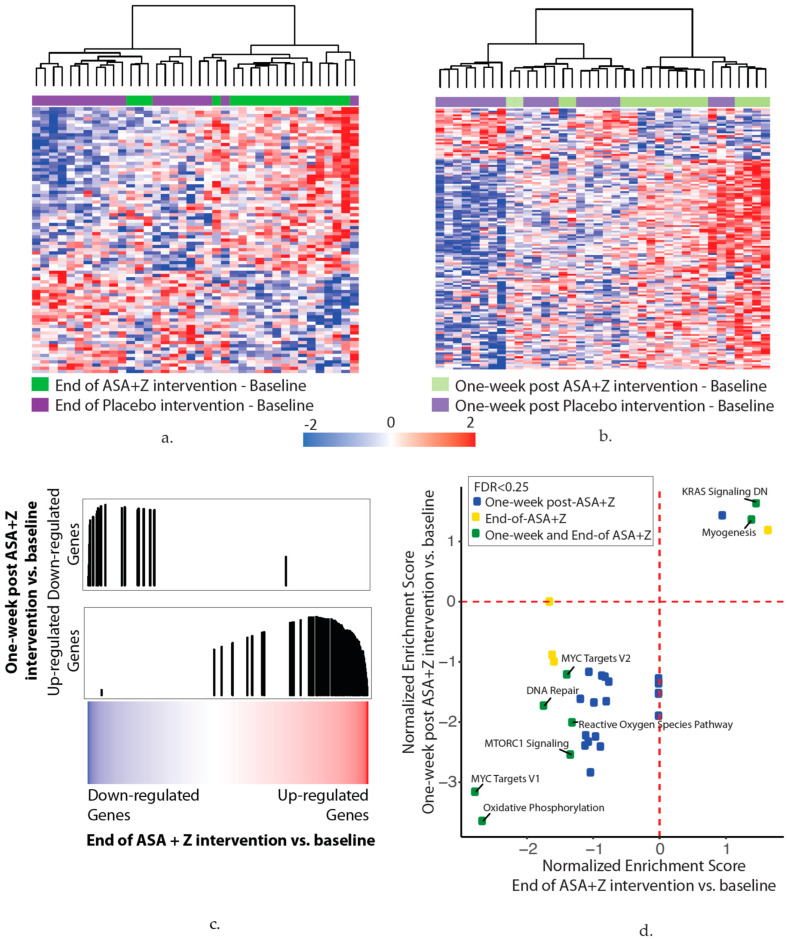
(**a**–**d**) Aspirin and zileuton treatment alter nasal gene expression.

**Table 1 cancers-14-02893-t001:** Baseline characteristics of the randomized participants; data is presented as mean ± standard deviation.

Variable	All (*n* = 63)	ASA + Zileuton (*n* = 31)	Placebo (*n* = 32)	*p* ^b^
Age	51.57 ± 9.85	49.55 ± 8.64	53.53 ± 10.67	0.11
BMI	28.23 ± 7.02	28.69 ± 6.60	27.78 ± 7.48	0.61
Packyears	34.67 ± 12.42	32.74 ± 12.52	36.53 ± 12.22	0.23
Male	36 (57.14%) ^a^	18 (58.06%)	18 (56.25%)	1.00
Female	27 (42.86%)	13 (41.94%)	14 (43.75%)	1.00
White	52 (82.54%)	28 (90.32%)	24 (75.00%)	0.15
Hispanic	7 (11.11%)	3 (9.68%)	4 (12.50%)	1.00

^a^ frequency (%). ^b^ derived from a two-sample *t*-test for continuous variables and Fisher’s exact tests for categorical variables.

**Table 2 cancers-14-02893-t002:** Baseline characteristics of the randomized participants; data is presented as mean ± standard deviation.

Variable	All (*n* = 63)	ASA + Zileuton (*n* = 31)	Placebo (*n* = 32)	*p* ^b^
Age	51.57 ± 9.85	49.55 ± 8.64	53.53 ± 10.67	0.11
BMI	28.23 ± 7.02	28.69 ± 6.60	27.78 ± 7.48	0.61
Packyears	34.67 ± 12.42	32.74 ± 12.52	36.53 ± 12.22	0.23
Male	36 (57.14%) ^a^	18 (58.06%)	18 (56.25%)	1.00
Female	27 (42.86%)	13 (41.94%)	14 (43.75%)	1.00
White	52 (82.54%)	28 (90.32%)	24 (75.00%)	0.15
Hispanic	7 (11.11%)	3 (9.68%)	4 (12.50%)	1.00

^a^ frequency (%). ^b^ derived from a two-sample *t*-test for continuous variables and Fisher’s exact tests for categorical variables.

**Table 3 cancers-14-02893-t003:** Summary of study intervention-related adverse events.

Adverse Event	Aspirin + Zileuton (*n* = 32)	Placebo (*n* = 31)
	Grades 1, 2 (%)	Grades 1, 2 (%)
Blood and Lymphatic System Disorders		
Anemia		1 (3.23)
Blood and lymphatic system disorders—Other-Hematocrit Low		1 (3.23)
Gastrointestinal disorders		
Abdominal pain		1 (3.23)
Diarrhea	1 (3.13)	
Dyspepsia	4 (12.5)	4 (12.9)
Nausea	1 (3.13)	
Stomach pain	1 (3.13)	1 (3.23)
Vomiting	1 (3.13)	
General disorders and administration site conditions		
Fatigue	1 (3.13)	
Investigations		
Alkaline phosphatase increased		1 (3.23)
Blood bilirubin increased	1 (3.13)	1 (3.23)
Investigations—Other-Macrocytosis	1 (3.13)	
Alanine aminotransferase increased	1 (3.13)	
Aspartate aminotransferase increased	1 (3.13)	1 (3.23)
Metabolism and nutrition disorders		
Dehydration		1 (3.23)
Nervous system disorders		
Dizziness	1 (3.13)	1 (3.23)
Headache	2 (6.25)	1 (3.23)
Psychiatric disorders		
Mania	1 (3.13)	
Skin and subcutaneous tissue disorders		
Rash maculo-papular	1 (3.13)	

**Table 4 cancers-14-02893-t004:** Baseline and changes in a smoking-associated gene-expression signature derived from nasal and bronchial brushings by Zhang [29]; data is presented as mean ± standard deviation.

All	Overall (*n* = 40)	ASA + Zileuton (*n* = 19)	Placebo (*n* = 21)	*p* ^a^
Baseline	0.31 ± 3.85	0.82 ± 4.00	−0.15 ± 3.75	0.43
Post Intervention (Int.)	0.37 ± 3.72	0.67 ± 4.20	0.11 ± 3.30	
1-week Post Int.	−0.64 ± 4.10	−0.03 ± 4.45	−1.19 ± 3.79	
Baseline—Post Int.	0.06 ± 2.55	−0.15 ± 2.89	0.26 ± 2.26	0.62
Baseline-1-week Post Int.	−0.73 ± 3.31	−0.24 ± 3.04	−1.16 ± 3.55	0.40
Female	*n* = 19	*n* = 8	*n* = 11	
Baseline	−0.26 ± 3.05	−1.26 ± 1.40	0.47 ± 3.74	0.18
Post Int.	−0.37 ± 3.10	−1.25 ± 3.08	0.27 ± 3.09	
1-week Post Int.	−1.14 ± 3.52	−2.08 ± 3.49	−0.45 ± 3.54	
Baseline—Post Int.	−0.11 ± 2.37	0.01 ± 2.93	−0.20 ± 2.01	0.86
Baseline-1-week Post Int.	−0.87 ± 2.98	−0.82 ± 3.57	−0.91 ± 2.66	0.95
Male	*n* = 21	*n* = 11	*n* = 10	
Baseline	0.83 ± 4.46	2.33 ± 4.62	−0.83 ± 3.83	0.11
Post Int.	1.05 ± 4.16	2.07 ± 4.48	−0.07 ± 3.68	
1-week Post Int.	−0.14 ± 4.66	1.61 ± 4.60	−2.10 ± 4.10	
Baseline—Post Int.	0.22 ± 2.76	−0.27 ± 2.99	0.76 ± 2.51	0.41
Baseline-1-week Post Int.	−0.58 ± 3.69	0.22 ± 2.66	−1.47 ± 4.58	0.33

^a^ derived from two-sample *t*-test.

**Table 5 cancers-14-02893-t005:** Baseline and changes in a squamous dysplasia-associated gene expression signature derived from bronchial brushings by Beane [22]; data is presented as mean ± standard deviation.

All	Overall (*n* = 40)	ASA + Zileuton (*n* = 19)	Placebo (*n* = 21)	*p* ^a^
Baseline	−0.14 ± 5.04 ^a^	0.27 ± 5.01	−0.51 ± 5.15	0.63
Post t Intervention (Int.)	−0.22 ± 4.48	−2.05 ± 3.50	1.43 ± 4.70	
1-week Post Int.	0.27 ± 6.29	−1.56 ± 3.56	1.91 ± 7.72	
Baseline—Post Int.	−0.08 ± 4.66	−2.31 ± 4.08	1.94 ± 4.28	<0.01
Baseline-1-week Post Int.	0.37 ± 5.90	−1.83 ± 5.37	2.35 ± 5.77	0.03
Female	*n* = 19	*n* = 8	*n*= 11	
Baseline	0.40 ± 5.12	0.15 ± 5.85	0.58 ± 4.82	0.86
Post Int.	−0.58 ± 4.21	−1.33 ± 4.18	−0.04 ± 4.35	
1-week Post Int.	0.07 ± 6.84	−2.68 ± 3.37	2.07 ± 8.11	
Baseline—Post Int.	−0.98 ± 3.58	−1.48 ± 4.82	−0.61 ± 2.56	0.62
Baseline-1-week Post Int.	−0.33 ± 5.96	−2.83 ± 6.67	1.49 ± 4.91	0.12
Male	*n* = 21	*n* = 11	*n* = 10	
Baseline	−0.63 ± 5.04	0.35 ± 4.62	−1.70 ± 5.50	0.36
Post Int.	0.10 ± 4.79	−2.57 ± 3.00	3.05 ± 4.75	
1-week Post Int.	0.46 ± 5.87	−0.66 ± 3.62	1.71 ± 7.71	
Baseline—Post Int.	0.73 ± 5.41	−2.93 ± 3.57	4.75 ± 4.09	<0.001
Baseline-1-week Post Int.	1.06 ± 5.92	−1.03 ± 4.27	3.39 ± 6.84	0.11

^a^ derived from two-sample *t*-test.

**Table 6 cancers-14-02893-t006:** Summary of the change in LTE4 levels by treatment group and by gender.

All	Overall (*n* = 43)	ASA + Zileuton (*n* = 21)	Placebo (*n* = 22)	*p* ^b^
Baseline	88.17 ± 60.65 ^a^	89.867 ± 68.35	86.55 ± 53.86	0.86
Post Int. ^c^	78.02 ± 90.79	32.25 ± 23.25	121.73 ± 108.97	
1-week Post Int.	103.09 ± 97.71	79.57 ± 52.60	125.53 ± 124.01	
Baseline—Post Int.	−10.14 ± 92.43	−57.62 ± 65.56	35.17 ± 92.67	<0.001
Baseline-1-week Post Int.	14.92 ± 85.67	−10.29 ± 60.02	38.98 ± 100.02	0.06
Female	*n*= 21	*n* = 10	*n* = 11	
Baseline	104.00 ± 65.78	114.90 ± 74.88	94.09 ± 58.15	0.48
Post Int.	104.07 ± 115.81	27.93 ± 22.90	173.28 ± 123.64	
1-week Post Int.	112.65 ± 126.20	73.49 ± 32.00	148.24 ± 167.35	
Baseline—Post Int.	0.07 ± 125.66	−86.97 ± 69.20	79.19 ± 113.19	<0.001
Baseline-1-week Post Int.	8.65 ± 111.39	−41.41 ± 65.73	54.15 ± 127.06	<0.05
Male	*n* = 22	*n* = 11	*n* = 11	
Baseline	73.06 ± 52.39	67.10 ± 55.66	79.01 ± 50.86	0.61
Post Int.	53.17 ± 48.85	36.16 ± 23.95	70.17 ± 61.65	
1-week Post Int.	93.96 ± 61.20	85.10 ± 67.39	102.82 ± 56.13	
Baseline—Post Int.	−19.89 ± 42.92	−30.94 ± 51.40	−8.84 ± 30.96	0.24
Baseline-1-week Post Int.	20.90 ± 52.76	18.00 ± 38.33	23.81 ± 66.01	0.80

^a^ mean ± standard deviation. ^b^ derived from two-sample *t*-test. ^c^ Intervention.

## Data Availability

The gene expression data has been deposited in the NCBI Gene expression Omnibus under accession GSE175616.

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
