# Peer review of "Clinical Study of Aspirin and Zileuton on Biomarkers of Tobacco-Related Carcinogenesis in Current Smokers"

_cancers, 2022, doi:10.3390/cancers14122893_

Round 1
Reviewer 1 Report
Dear authors,
Congratulations on the performed work, you clearly worked hard for preparing the manuscript, demonstrating that low dose ASA and zileuton compared to placebo modulate a bronchial squamous dysplasia gene signature, even though it has no significant effects on other known carcinogenesis gene signatures in the nasal epithelium of heavy smokers. Your work provides the basis of future research of this topic and might lead to the implementation of prevention strategies for tobacco-related lung cancer in smokers.
I have only a minor suggestion to add to the quality of your work, in a further study/analysis. It would be interesting to also apply a quality-of-life assessment, to investigate how the intervention (ASA-zileuton vs. placebo) can improve/decrease the QoL of the participants.
Nevertheless, the results are clearly presented, the statistical methods easy to follow and the discussion well exposed.
Kind regards,
Author Response
Reviewer 1:
I have only a minor suggestion to add to the quality of your work, in a further study/analysis. It would be interesting to also apply a quality-of-life assessment, to investigate how the intervention (ASA-zileuton vs. placebo) can improve/decrease the QoL of the participants.
Author reply:
This valuable comment has been addressed in the following paragraph and that addresses study weakness (lines 738-741):
A weakness in the study design is the lack of quality of life (QOL) endpoints such that the impact of this ASA plus zileuton strategy on QOL can be assessed and used to guide similar chemoprevention strategies with the goal of, at the most, modest impact of QOL in an at-risk but otherwise healthy population.
Reviewer 2 Report
Thank you for the chance you gave me to read this interesting study entitled “Clinical study of aspirin and zileuton on biomarkers of tobacco-related carcinogenesis in current smokers” by Garlandet al. In this study, the authors present the results of a randomized, double-blinded study of low dose aspirin plus zileuton vs. double placebo in current smokers in order to assess their preventive role on lung cancer development by modulating effects on nasal epithelium gene expression and arachidonic acid metabolism. This topic is clinically oriented and has great importance. The study is well-designed and organized as well as well presented. I think that this study in the current form does satisfy the appropriate criteria for publication in your journal, however, some minor points should be treated before being suitable for publication.
Some of them are:
- Abstract needs to be improved describing more clearly the results.
- Abbreviations should be expanded at first mention e.g. LTB line 81.
- Lines 106-109: Please, clarify the meaning of the sentence. It needs to be rewritten.
- The authors should explain why a “12-week intervention” with the combination was chosen since usually longer periods of administration of chemoprevention are used. Although they provide some points regarding this issue at discussion, I think that they should interpret more this weak point, which maybe influenced significantly the results of the current study.
- A paragraph with weak points of the study at discussion is needed.
Author Response
Reviewer 2 comments are very valuable and are responded to below by the author:
- Abstract needs to be improved describing more clearly the results.
The abstract has been revised in order to more clearly inform the reader as to scientific findings. Please see the track changes version for what has been added and the final version is as follows:
Abstract: A chemopreventive effect of aspirin and other non-steroidal anti-inflammatory drugs (NSAIDs) on lung cancer risk is supported by epidemiologic and preclinical studies. Zileuton, a 5-lipoxygenase inhibitor, has additive activity with NSAIDs against tobacco carcinogenesis in preclinical models. We hypothesized cyclooxygenase plus 5-lipoxygenase inhibition would be more effective than placebo in modulating nasal epithelium gene signatures of tobacco exposure and lung cancer. We conducted a randomized, double-blinded study of low dose aspirin plus zileuton vs. double placebo in current smokers to compare modulating effects on nasal gene expression and arachidonic acid metabolism. Sixty-three participants took aspirin 81 mg daily plus zileuton (Zyflo CR) 600 mg BID or placebos for 12 weeks. Nasal brushes from baseline, end-of-intervention, and one-week post intervention were profiled via microarray. Aspirin plus zilueton had minimal effects modulating nasal or bronchial gene expression signatures of smoking, lung cancer and COPD, but favorably modulated a bronchial gene expression signature of squamous dysplasia. Aspirin plus zileuton suppressed urinary leukotriene but not prostaglandin E2, suggesting shunting through the cyclooxygenase pathway when combined with 5-lipoxygenase inhibition. Continued investigation of leukotriene inhibitors is needed to confirm these findings, understand long-term effects on the airway epithelium, and identify the safest, optimally dosed agents.
2. Abbreviations should be expanded at first mention e.g. LTB line 81.
Abbreviations have been expanded for those in lines 133 and 134.
3. Lines 106-109: Please, clarify the meaning of the sentence. It needs to be rewritten.
The following revised sentences (now lines 135-148) are as follows and for better clarity, the lines beginning with "In preclinical studies..." have been separated as into their own paragraph:
Additional supporting evidence for 5-LOX as a chemoprevention target is evidence that mRNA for 5-LOX and 5-LO activating protein (FLAP), which is required for activation of 5-LOX, is expressed in lung cancer cell lines and that mRNA for 5-LOX is expressed in lung cancer tissues [11].
In preclinical studies, 5-LOX inhibitors including zileuton have inhibitory activity in a number of lung cancer models [12,13]. In an A/J mouse lung chemoprevention model using vinyl carbamate induction, the administration of zileuton 1200 mg/kg in the diet (equivalent to a clinically relevant human dose) starting 2 weeks after carcinogen exposure caused a significant reduction in tumor multiplicity (24 % at 13 weeks and 28% at 43 weeks) and reduced the size of lung tumors [14].
4. The authors should explain why a “12-week intervention” with the combination was chosen since usually longer periods of administration of chemoprevention are used. Although they provide some points regarding this issue at discussion, I think that they should interpret more this weak point, which maybe influenced significantly the results of the current study.
More supporting data for length of intervention has been added (lines 703-707) to the paragraph and as follows:
The optimal duration of combined COX and 5-LOX inhibitors to modulate tobacco- and lung cancer-related gene expression in the nasal epithelium as a surrogate for the respiratory epithelium is unknown. The study’s relatively short (12 week) intervention is broadly consistent with a number of preclinical biomarker studies of NSAIDs and other classes of chemoprevention agents in murine models of cigarette smoke exposure [44,45]. While murine models may not adequately represent the pharmacologic modulation of human chronically smoke exposed respiratory epithelium as murine models require high doses of both carcinogen and chemopreventive agent to detect an effect in a short period of time, our previously reported study of low dose ASA in a similarly designed study of 12 week intervention showed modulation of gene expression signatures over this duration of intervention [29]. Future studies that include cohorts exposed to longer durations of intervention could provide valuable data on optimal dosing strategies.
In addition, a comment about the study design limitation of a single study duration rather than including a longer duration such as 6 months or longer has been added to the paragraph addressing study weak points (see below under comment #5.)
5. A paragraph with weak points of the study at discussion is needed.
The following paragraph to address weak points of the study (lines 738-745):
A weakness in the study design is the lack of quality of life (QOL) endpoints, which could elucidate the impact of this ASA plus zileuton strategy on QOL and thus guide similar chemoprevention strategies with the goal of, at the most, modest impact on QOL in an at-risk but otherwise healthy population. A second weakness in the study design is the inability to better inform optimal duration of intervention to modulate the study endpoints, given that the single duration intervention was short (3 months). Additional cohorts utilizing longer durations of intervention ( i.e., 6- or 12- months) would have strengthened the study.